# Combining HF rTMS over the Left DLPFC with Concurrent Cognitive Activity for the Offline Modulation of Working Memory in Healthy Volunteers: A Proof-of-Concept Study

**DOI:** 10.3390/brainsci10020083

**Published:** 2020-02-04

**Authors:** Ilya Bakulin, Alfiia Zabirova, Dmitry Lagoda, Alexandra Poydasheva, Anastasiia Cherkasova, Nikolay Pavlov, Peter Kopnin, Dmitry Sinitsyn, Elena Kremneva, Maxim Fedorov, Elena Gnedovskaya, Natalia Suponeva, Michael Piradov

**Affiliations:** 1Research Center of Neurology, Volokolamskoe Shosse, 80, Moscow 125367, Russia; alfijasabirowa@gmail.com (A.Z.); dmitrylagoda.doc@gmail.com (D.L.); alexandra.poydasheva@gmail.com (A.P.); cherka.sova@mail.ru (A.C.); nickvolvap@gmail.com (N.P.); kopnin.p@gmail.com (P.K.); d_sinitsyn@mail.ru (D.S.); moomin10j@mail.ru (E.K.); gnedovskaya@mail.ru (E.G.); nasu2709@mail.ru (N.S.); mpi711@gmail.com (M.P.); 2Skolkovo Institute of Science and Technology, Bolshoy Boulevard, 30, bld. 1, Territory of Innovation Center «Skolkovo», Moscow 121205, Russia; M.Fedorov@skoltech.ru

**Keywords:** working memory, non-invasive brain stimulation, transcranial magnetic stimulation, neuromodulation, cognitive function, cognitive training, N-back task, dorsolateral prefrontal cortex, cognitive enhancement

## Abstract

It has been proposed that the effectiveness of non-invasive brain stimulation (NIBS) as a cognitive enhancement technique may be enhanced by combining the stimulation with concurrent cognitive activity. However, the benefits of such a combination in comparison to protocols without ongoing cognitive activity have not yet been studied. In the present study, we investigate the effects of fMRI-guided high-frequency repetitive transcranial magnetic stimulation (HF rTMS) over the left dorsolateral prefrontal cortex (DLPFC) on working memory (WM) in healthy volunteers, using an n-back task with spatial and verbal stimuli and a spatial span task. In two combined protocols (TMS + WM + (maintenance) and TMS + WM + (rest)) trains of stimuli were applied in the maintenance and rest periods of the modified Sternberg task, respectively. We compared them to HF rTMS without a cognitive load (TMS + WM −) and control stimulation (TMS − WM + (maintenance)). No serious adverse effects appeared in this study. Among all protocols, significant effects on WM were shown only for the TMS + WM − with oppositely directed influences of this protocol on storage and manipulation in spatial WM. Moreover, there was a significant difference between the effects of TMS + WM − and TMS + WM + (maintenance), suggesting that simultaneous cognitive activity does not necessarily lead to an increase in TMS effects.

## 1. Introduction

Working memory (WM) can be defined as a combination of temporary storage of task-relevant information and processing upon it. WM performance has been shown to be associated with complex cognitive functions such as learning ability and fluid reasoning [1,2]. A decline of WM performance accompanies a range of neurologic and psychiatric diseases (e.g., neurodegenerative dementias, schizophrenia, depression and many others [3]) as well as healthy aging [4].

Dorsolateral prefrontal cortex (DLPFC) is one of the brain areas playing a key role in WM processes [5,6], which was initially determined in invasive neurophysiologic studies performed on non-human primates [7]. Further human studies using functional magnetic resonance imaging (fMRI) have also confirmed the activation of PFC as a part of WM-related neural networks during WM tasks [8,9,10]. Moreover, the activation of DLPFC during the WM task can be load-dependent, providing additional evidence for its role in WM processes [11].

Due to its important role in WM processes, DLPFC is intensively investigated as a possible target for the treatment of WM impairments using non-invasive brain stimulation (NIBS) techniques, such as repetitive transcranial magnetic stimulation (rTMS). TMS is based on the ability of a magnetic field applied to the head to induce electric current in the underlying tissues, leading to depolarization of neural axons and bodies [12]. The use of focal coils and stereotactic neuronavigation systems (navigated TMS) for their positioning enables precise stimulation of small brain areas [13,14]. The combination of navigated TMS with functional neuroimaging data can be used for more accurate determination of individualized stimulation targets in the brain areas specifically activated during a task [15].

Along with the “short-term” effects of TMS on ongoing neural activity during stimulation (online-effects), many rTMS protocols can also induce long-term effects, lasting beyond the stimulation session (offline-effects), due to their ability to influence the processes of synaptic plasticity [16,17]. The investigation of rTMS offline-effects on cognitive functions is very important in the context of its use as a therapeutic technique. TMS over the left DLPFC has already been used both for investigation of its causal role in WM processes in healthy volunteers [18,19,20,21,22] and as a method of WM improvement in patients with WM impairments [23,24]. In a meta-analysis of studies using high frequency (HF) rTMS over the left DLPFC, a significant improvement of WM accuracy and reaction times measured by n-back task has been confirmed [25]. However, the authors emphasized a notable inhomogeneity of the analyzed studies due to the use of different rTMS protocols, small sample sizes in the studies, and the individual variability of the responses to TMS [25]. 

A novel approach to the use of NIBS for improvement of cognitive functions, including WM, is the combination of NIBS and simultaneous performance of a cognitive task, based on the fact that the effects of NIBS depend on the brain activity state during stimulation [26]. State-dependency of TMS effects was firstly established in studies with online TMS protocols. In these studies, TMS was applied in order to interrupt an ongoing neural activity in the stimulated area by increasing task-unrelated (noisy) neural activity. In this case, TMS induced so-called “virtual lesions”, which can be defined as a reduced task performance in cognitive studies [12]. State-dependency of TMS effects should also be taken into account in case of its long-term effects. According to this conception, NIBS effects are considered as an interaction between the stimulation itself and the underlying state of a brain region or a network [26]. It is proposed that simultaneous performance of a cognitive task during stimulation can potentially enhance the effects compared to stimulation alone [26]. Consistent with this hypothesis are the results of a recent study that showed an increase in cognitive function in patients with Alzheimer’s disease after rTMS in combination with simultaneous performance of cognitive tasks [27]. However, in this study, combined rTMS protocol was not compared to a protocol without cognitive activity; therefore, it provides no evidence about the ability of concurrent cognitive activity to increase the effects of NIBS.

The activity state of a target brain area is changing during a cognitive task, which, considering the state-dependency of NIBS effects, might influence the general effects of the stimulation. In WM studies, the changes of DLPFC activity were confirmed using so-called delayed-response tasks, which consist of three separate stages with a fixed duration: encoding, maintenance, and retrieval. First, Fuster and colleagues have shown in their studies performed on non-human primates that neurons in DLPFC are activated not only during a presentation of a task stimulus (food in a box) but also in the maintenance period of the WM task [7]. This ability of the neurons in DLPFC to be activated in the absence of the stimulus is called persistent neural activity or delay-period activity [5,28]. An example of a delayed-response task, frequently used in human studies, is the Sternberg task (delayed item recognition task) [29]. It has been shown in fMRI studies that DLPFC activity changes during the Sternberg task, becoming the most prominent at maintenance and retrieval stages [11,30,31]. 

A relatively uninvestigated problem that might contribute to the effects of stimulation combined with cognitive activity is the temporal synchronization of NIBS protocols with complex tasks. The problem of temporal synchronization is particularly relevant for such NIBS techniques as high-frequency rTMS (HF rTMS) or intermittent theta-burst stimulation (iTBS), where the stimuli are applied in short trains lasting several seconds separated by intertrain intervals (ITIs) [16]. The temporal pattern of DLPFC activation could influence the effects of stimulation and should be taken into consideration in the interpretation of the effects of patterned rTMS protocols on cognitive functions [32]. A combination of the Sternberg task with HF rTMS can be used to enable a relatively precise and reproducible temporal synchronization of TMS trains with ongoing cognitive activity, which is particularly valuable for investigation of WM in cognitive neuroscience: it might lead to more consistent stimulation effects and enables more accurate determination of a functional role of stimulated cortical areas in WM processes.

In a recent online-HF rTMS study, two different online-protocols (trains of stimuli applied in the maintenance and before encoding periods of a delayed-response WM task) have been applied to a DLPFC area activated during the same WM task according to fMRI data, and no significant difference between these protocols has been shown [33]. However, to our best knowledge, there are no studies assessing offline-effects (i.e., effects of stimulation beyond the duration of a session) of HF rTMS protocols in combination with a cognitive task on WM performance in comparison with effects of rTMS without a cognitive load.

The aim of the present proof-of-concept study is to determine if concurrent cognitive activity in different combinations with individual fMRI-guided HF rTMS over the left DLPFC has an influence on its offline-effects on WM performance in comparison to rTMS alone. Therefore, we developed two combined protocols of HF rTMS with trains of stimuli applied during maintenance and rest periods of the Sternberg task, and compared their effects to the HF rTMS without cognitive load and control HF rTMS over the vertex with trains of stimuli applied during maintenance.

## 2. Materials and Methods

### 2.1. Participants

Twelve healthy volunteers meeting the inclusion criteria listed below were initially enrolled in the study. All participants provided written informed consent about their enrollment in the study. The study was conducted in accordance with the Declaration of Helsinki. The study was approved by the Local Ethical Committee of Research Center of Neurology (protocol 9-3/17, 30 August 2017). History of neurological and psychiatric diseases and use of drugs acting on the central neural system was taken by a physician before the study. In order to exclude epileptiform activity on EEG, a routine EEG with activation procedures (photostimulation and hyperventilation) was performed (10–20 system, total duration of 15 min).

Inclusion criteria for participation in this study were as follows:age of 18–55 years;normal or corrected-to-normal vision;right-handedness according to Edinburgh Handedness Inventory [34];provided written informed consent about participation in the study.

Exclusion criteria consisted ofMRI and TMS contraindications, e.g., implanted cardiac devices (pacemakers, intracardial catheters), electronic pumps, ear implants, magnetic clips and stents, other magnetic implants, postsurgical devices or foreign bodies, claustrophobia, pregnancy, and others according to guidelines [35,36];epileptiform discharges on EEG;history of neurologic and psychiatric diseases;severe chronic diseases;intake of drugs acting on central nervous system.

### 2.2. Assessment of rTMS Safety and Tolerability

Safety and tolerability of HF rTMS protocols were assessed using standardized questionnaires for self-report of adverse effects (AEs) during each session and within 24 h after stimulation. The questionnaires for assessment of AEs during a session included the following sections: pain (intensity, location and descriptors), non-painful unpleasant sensations (e.g., muscle contractions or a burning sensation in the stimulated area), presence of nausea, vertigo, drowsiness, concentration difficulties, and other AEs. At the end of the questionnaire, participants also answered the question regarding their willingness to continue stimulation depending on their individual tolerability of AEs. The questionnaires for assessment of AEs within 24 h after stimulation included detailed assessment of headache (intensity, time of onset, duration, descriptors, use of analgesics for pain relief, accompanying symptoms, influence on daily activity), pain in the neck and non-painful AEs (hearing loss or tinnitus, concentration difficulties, mood changes and others), as well as other AEs. 

### 2.3. Experimental Design

At the first visit, a screening EEG and task-fMRI described below were performed for all participants. In the present study, a cross-over design was used, therefore at visits 2–5, all participants received different protocols of fMRI-guided navigated HF rTMS in randomized order using a random number approach. For the assessment of rTMS effects, they performed cognitive tests before and 10 min after the stimulation session. Visits 2–5 were separated by an interval of no less than 1 week for wash-out of rTMS effects.

#### 2.3.1. fMRI Task

In order to determine an individual target for further navigated rTMS in our study, we used fMRI with a verbal WM task based on the modified Sternberg task (delayed stimulus recognition task), adapted from Narayanan et al. (2005) [11]. The task consisted of four phases: encoding, maintenance, retrieval, and rest (Figure 1). At the encoding stage (3 s), the stimulus (7 random latin consonants, projected on the display of MRI compatible monitor) was shown. During maintenance, the participants had to actively remember the stimulus for 9 s with no visual stimulation (black screen). At the retrieval stage (3 s), a single latin letter was presented on the screen and participants compared it with the 7 consonants presented during encoding. The participants could see the information on display through a folding mirror placed on the head coil. They were instructed to press a button on the right hand-held grip if the letter shown in retrieval matched the previously remembered sequence, and in case of non-matching stimulus, they had to press a button on the left hand-held grip. The matching and non-matching stimuli were pseudorandomized during each fMRI with the 1:1 ratio between them. The rest period had a duration of 15 s. The task was repeated 24 times with different stimuli for a total duration of 12 min. During the scanning, all participants used headphones to minimize the scanner noise.

For the acquisition of MRI data, a 3T scanner Magnetom Verio (Siemens, Germany) was used. Anatomic structural data were acquired using 3D-T1-gradient echo sequence (T1-MPR) and consisted of 176 sagittal slices (TR = 1900 ms, TE = 2.47 ms, slice thickness= 1.0 mm, voxel size 1.0 × 1.0 × 1.0 mm^3^, FOV = 250 mm). Functional data were acquired using an MRI T2*-gradient echo sequence (TR = 3000 ms, TE = 30 ms, slice thickness = 3 mm, voxel size = 3.0 × 3.0 × 3.0 mm^3^, FOV = 192 mm, number of slices = 36). The first 5 functional images were excluded from the analysis to achieve dynamic magnetic equilibrium.

#### 2.3.2. fMRI Pre-Processing and Analysis

Image pre-processing and individual statistical analysis to create fMRI masks used for guided rTMS were performed using SPM12 (Statistical Parametric Mapping; Functional Imaging Laboratory, Wellcome Department of Imaging Neuroscience, Institute of Neurology, London, UK; http://www.fil.ion.ucl.ac.uk/spm/) based on MatlabR2010a (Mathworks, Natick, MA, USA). Pre-processing of fMRI data included the following stages: realignment of functional images, co-registration of anatomic and functional data, normalization to an MNI template, and spatial smoothing with a Gaussian kernel of 8 mm FWHM. Individual BOLD signals were modelled by a GLM with three active conditions (encoding, maintenance, and retrieval), described as boxcar functions, convolved with the canonical hemodynamic response function. Six head motion parameters were included in the model as nuisance regressors. An area within the left DLPFC showing maximal significant positive activation during maintenance according to task-fMRI data was visually determined as the target for active stimulation.

#### 2.3.3. Cognitive Tests

WM performance was measured before and 10 min after each rTMS session using n-back task (Psychology Experiment Building Language (PEBL) Battery [37]), and spatial span task (SSP) (Cambridge Neuropsychological Test Automated Battery (CANTAB), UK, https://www.cambridgecognition.com/cantab/). Before the SSP task, a motor screening task was performed to familiarize the participants with the computerized procedure and to ensure that they had no impairments in their ability to react to the stimuli. Participants performed tests sitting in a comfortable chair in front of a PC monitor with a touch-sensitive screen, where stimuli were presented. The influence of distractors during testing was minimal. 

In our study, we used the n-back task [38] with simultaneous presentation of 2 types of visual stimuli: verbal (latin consonants) and spatial (1 of 8 possible positions of a square in a 3 × 3 grid except for the central square, where the letter was located). In this task, participants had to compare test stimuli currently appearing on the screen with the stimuli presented n-steps back (*n* = 1, 2, 3). They were instructed to press the left Shift key in case of matching verbal stimuli and the right Shift in case of matching spatial stimuli. Each n-back test session included training with separate presentations of verbal stimuli (*n* = 1, 2), spatial stimuli (*n* = 1) and simultaneous presentation of verbal and spatial stimuli (*n* = 1). After the training, participants performed a scored test with a simultaneous presentation of verbal and spatial stimuli (*n* = 1, 2, 3 with 21, 22 and 23 stimuli, respectively). In each scored session, 6 verbal and 6 spatial stimuli were the targets. For each session, total amounts of hits and false alarms were recorded.

For the SSP task, we used a high functioning reverse mode. In this task, a number of white boxes were presented on the screen. The boxes were colored one-by-one in a random order. After an auditive signal, the participant had to reproduce the location of colored boxes in reverse order. The number of boxes increased after each right answer. In case of false answers, the same number of boxes could be repeated up to 3 times. The task was finished after 3 false answers. A training session consisting of 2 trials with 4 boxes presented and a scored test began with 4 boxes presented, followed by trials with 5, 6, 7, 8, 9, 10, 12, 15, and 18 boxes. For each session, SSP reverse span length (the number of boxes in the longest successfully remembered sequence) was recorded.

#### 2.3.4. Stimulation Protocols

For our fMRI-guided navigated HF rTMS, we used an NBS eXimia Nexstim stimulator (Nexstim, Finland) with a bi-pulse focal (figure-of-eight) coil. Stimulation intensity was set at 100% of resting motor threshold (rMT) of the right abductor pollicis brevis (APB) muscle. RMT was measured in the hot spot of APB cortical representation and determined as minimal stimulation intensity inducing MEPs with the peak-to-peak amplitude larger than 50 µV in more than 5 of 10 trials [13].

In this study, we compared 4 different HF rTMS protocols: 3 active stimulation protocols, in which rTMS was applied over the individually determined target within the left DLPFC, and a control protocol, for which rTMS of the vertex was used (Figure 2). Trains of HF rTMS with a frequency of 10 Hz had a duration of 4 seconds and ITIs of 26 s between them. Each session consisted of 40 trains (1600 stimuli) and had a total duration of 20 min. In order to investigate the state-dependency of rTMS effects, stimulation was combined with the modified Sternberg task analogous to the task used in our fMRI paradigm. The task and HF rTMS were synchronized using a Chronos response and stimulus device and E-Prime 3.0 software (Psychology Software Tools, Sharpsburg, MD, USA). Trains of stimuli were applied either during active maintenance of the presented stimuli (TMS + WM + (maintenance): stimulation from the 3rd until the 6th second of the maintenance period, inclusively) or during the rest period between the presentations of the Sternberg task (TMS + WM + (rest): from the 6th until the 9th second of the rest period, inclusively). The protocol without ongoing cognitive activity (TMS + WM −) consisted of 20 min of HF rTMS alone, as described above. As a control condition (TMS − WM + (maintenance)), we used a protocol analogous to TMS + WM + (maintenance), except that the vertex was used as a stimulation target (Figure 2).

### 2.4. Statistical Analysis

Statistical analysis was performed using the SPSS Statistics software package, version 23 (IBM, USA) and MatlabR2010a (Mathworks, Natick, MA, USA). N-back performance was evaluated using d prime sensitivity index (d′) [39]. The calculation of this index is based on 2 types of answers: hits (correct response to matching stimulus) and false alarms (incorrect response to non-matching stimulus). The d′ scores were calculated individually as
d′ = *Z*(hit rate) − *Z*(false alarm rate)(1)
where *Z* represents a transformation of the two distributions in order to calculate the difference between two rates. In this method, d’ can be considered as the normalized distance between the probability distributions of signal and noise and noise alone. The rate is highest when the participant has a maximal rate of hits and a minimal rate of false alarms confirming his ability to discriminate matching and non-matching stimuli during the task. A high rate of correct responses in the n-back task with n = 1 leads to an extremely high probability of the so-called ceiling effect in such an easy-level task, therefore in our study, d’ scores were not calculated for this condition and the data were not included in the analysis. Performance in the SSP task was assessed as the number of squares in the longest correctly reproduced sequence (reverse span length).

Because of the non-Gaussian distribution of test results (confirmed with Kolmogorov–Smirnov and chi-square tests; *p* < 0.05), non-parametric statistical tests were used for the analysis. At the first stage, we compared baseline performances before each protocol to exclude baseline inhomogeneity of test scores using Friedman’s ANOVA. We compared test performance before and after one single session of each protocol using the Wilcoxon test. Fisher’s *p*-value synthesis was additionally performed to determine if the main effect of each protocol on all cognitive tests was significant. However, it should be mentioned that due to the cross-over design of our study, the application of *p*-value synthesis has a limitation because the compared samples are not independent.

For statistical comparison between protocols, we calculated and compared deltas for each protocol applying Friedman’s ANOVA. Spearman rank correlation was calculated to investigate the possible correlations between the participants’ scores in n-back and SSP tasks. In order to determine a possible impact of the learning effect on changes in WM performance, we also compared the test scores before stimulation and the differences between after and before stimulation (delta) for each visit in a chronological order using Friedman’s ANOVA. All data are shown in the following format: median (Q1; Q3).

## 3. Results

### 3.1. Participants

A total of 12 right-handed healthy volunteers (4 males; age 22–31 years), who met the inclusion criteria, were enrolled in the study. After two sessions of stimulation, one person rejected participation in the study due to logistic problems. The final number of participants included in the following analysis was 11. The stimulation target was determined within the left DLPFC visually as the region with the maximal activation according to individual fMRI data (Figure 3).

### 3.2. Behavioral Results

The d’ scores of the *n*-back task with verbal and spatial stimuli (*n* = 2,3) and SSP were analyzed (Table 1, Appendix A). Baseline performances before each protocol were compared using Friedman’s ANOVA and no significant differences were found between protocols in all tests (for *n*-back with *n* = 2, *p* = 0.82 for verbal and *p* = 0.29 for spatial stimuli; with *n* = 3, *p* = 0.46 for verbal and *p* = 0.08 for spatial stimuli; for SSP *p* = 0.89).

### 3.3. rTMS Influence on Task Performance

In order to determine the effect of different HF rTMS protocols on both verbal and spatial WM performance, we compared test scores before and after each session using a non-parametric Wilcoxon test. Then Fisher’s *p*-value synthesis was performed to determine if there was an effect on any of the cognitive tests for each protocol. A significant increase in SSP scores (*p* = 0.040) and a significant decrease in high-load (*n* = 3) n-back tasks with spatial stimuli (*p* = 0.045) were found after TMS + WM −. Moreover, a significant overall effect on all tests, according to the results of *p*-value synthesis, was shown only for this protocol (*p* = 0.03). No significant differences in performance after other protocols were shown in the present study (Table 2).

After the assessment of WM performance differences following application of each HF rTMS protocol, we compared changes in test scores measured as the difference between test scores before and after stimulation (delta) using non-parametric Friedman’s ANOVA. A significant difference between 4 protocols was found for the high-load (*n* = 3) n-back task with spatial stimuli (*p* = 0.043). No other significant differences were shown (for medium-load (*n* = 2) n-back task with verbal stimuli *p* = 0.537, with spatial stimuli *p* = 0.317, for high-load n-back task with verbal stimuli *p* = 0.219, for SSP *p* = 0.121).

For the high-load n-back task with spatial stimuli, a post-hoc analysis using the Wilcoxon test with Bonferroni correction for multiple comparisons was performed. A significant difference was found for performance changes after TMS + WM + (maintenance) and TMS + WM − (*p* = 0.048). No other significant differences after multiple paired between-group comparisons were found (Table 3).

For two test scores (high-load n-back task with spatial stimuli and SSP) showing a significant difference between performance after and before TMS + WM −, a Spearman rank correlation coefficient was calculated in order to determine a possible correlation between their scores. However, no significant correlation was found (Spearman r = −0.231 (*p* = 0.495)).

In order to determine learning effects on test performance, we also compared test scores before each day of stimulation session using Friedman’s ANOVA for multiple dependent variables. There were no significant differences and, therefore, no significant learning effect for all n-back tests (for *n* = 2 with verbal stimuli *p* = 0.644; with spatial stimuli *p* = 0.139; for *n* = 3 with verbal stimuli *p* = 0.370; with spatial stimuli *p* = 0.799). There was a significant learning effect in the SSP task (*p* = 0.016). However, this learning effect had no significant influence on the differences in test performance in all tests measured by deltas (for n-back test with *n* = 2 and verbal stimuli *p* = 0.054, spatial stimuli *p* = 0.154; for n-back with *n* = 3 and verbal stimuli *p* = 0.559, spatial stimuli *p* = 0.965, for SSP *p* = 0.139). Moreover, protocols were applied in a randomized order, diminishing the impact of learning effect on WM performance.

### 3.4. TMS Tolerability

No serious AEs leading to the cessation of participation were reported in our study. AEs were reported in 26 sessions (59.1% of all sessions, total number of sessions = 44) and had mild (*n* = 22) or moderate (*n* = 4) intensity with no influence on the participants’ willingness to continue the stimulation. There was no significant difference between frequencies of AEs during the stimulation over left DLPFC compared to the control stimulation over vertex (20 of 33 sessions and 6 of 11 sessions, respectively; Fisher’s exact test *p* = 0.738).

In our study, non-painful AEs were the most frequent AEs reported during stimulation. They included drowsiness (*n* = 16), muscular contractions (*n* = 9), and concentration difficulties (*n* = 4). Notably, muscular contractions in the stimulated area appeared only during DLPFC stimulation. In one case, the participant reported lacrimation that accompanied muscular contractions in the stimulated area. Headache during stimulation occurred in 9 sessions (7 sessions of DLPFC stimulation and 2 sessions of vertex stimulation) and was mostly described as throbbing or stabbing. Pressing pain in the right fronto-temporo-parietal area was reported once after the vertex stimulation. Otherwise, the headache was localized mostly near to the stimulated area (frontal or fronto-temporal for DLPFC and bilateral parietal for vertex) and had mild intensity (2–4 points on the visual analogous scale (VAS)).

AEs occurring within 24 h were reported in 5 cases (11.4% of all sessions). They included headache (*n* = 4), concentration difficulties (*n* = 2), and drowsiness (*n* = 1). Headache was most often described as dull, occurred within the first hour after stimulation and had moderate intensity (2–6 points on VAS). It resolved spontaneously or after intake of NSAIDs. Headache occurring after the DLPFC stimulation (2 cases) was localized in the fronto-temporal area, had an intensity of 2–4 points, and in one case, was accompanied by nausea and vertigo lasting 15 min after the stimulation. Headache after the vertex stimulation had, in one case, an intensity of 4 points and a bilateral fronto-temporal localization, lasting 1–2 h. In another case of headache after the vertex stimulation, the headache was diffuse, had moderate intensity (6 points, with maximal intensity within 1–2 h after the stimulation) and was accompanied by lacrimation, a dull sensation in the left orbita, neck stiffness, and had a total duration of more than 24 h after the stimulation. Concentration difficulties appeared both after the DLPFC and the vertex stimulation, and in one case, accompanied the headache.

## 4. Discussion

In this proof-of-concept cross-over study, we investigated if different combinations of fMRI-guided navigated HF rTMS over the left DLPFC with a cognitive task might influence the effects of a single rTMS session on different components of WM compared to stimulation alone, as well as to control stimulation. We studied two combined HF rTMS protocols with trains of stimuli applied during maintenance and rest periods of the modified Sternberg task, a protocol without a cognitive load and control rTMS during maintenance. However, we have not shown any data confirming any significant effects of combined protocols. Moreover, a significant effect was shown only after HF rTMS without a cognitive load on spatial WM. This effect was dependent on a task type: an increase in SSP task performance, a storage-based WM task, and a decrease in the score of high-load n-back task with spatial stimuli, in which manipulation plays a crucial role [9,40,41] were found. Therefore, an oppositely directed influence of HF rTMS over the left DLPFC on different WM processes (storage and manipulation, respectively) can be suggested. However, we did not observe any significant differences between the effects of active and control protocols. Thus, the state-dependency of HF rTMS effects as well as the source of its oppositely directed influence on different components of WM require further investigation. Additionally, we confirm that the HF rTMS protocols used in our study are safe and well-tolerated by healthy volunteers and can be used in further studies.

In order to diminish the interindividual variability of the response to NIBS in our study, we used an individual-fMRI-guided targeting of the left DLPFC. High variability is one of the most prominent problems of NIBS studies, frequently leading to moderate or absent effects of stimulation. In cognitive studies, different activity states of investigated brain areas during a cognitive task might be an additional source of such variability. The trains of stimuli applied in our protocols matched the period of maximal DLPFC activity (TMS + WM + (maintenance)) and the period of decrease in DLPFC activation (TMS + WM + (rest)), as has been shown in fMRI studies [11], which is generally consistent with the hypothesis of the role of persistent neural activity in DLPFC in WM [5]. This approach allowed us to compare the effects of the stimulation during two different activity states of DLPFC. 

State-dependency of rTMS effects is one of the most controversial topics limiting the use of rTMS both in cognitive studies and as a therapeutic approach. In contrast to other studies, assessing online-effects of the stimulation over DLPFC on the test performance immediately during a stimulation session (a standard approach, firstly applied in “virtual lesion” studies), we investigated if combination of HF rTMS with a cognitive task had an influence on its offline-effects lasting beyond a stimulation session. To our best knowledge, the offline-approach was previously used in WM studies only for the investigation of stimulation without the simultaneous performance of a cognitive task. However, our data did not provide significant evidence for a non-zero effect on WM performance of any of the combined protocols used in our study. Furthermore, a significant effect of the protocol without cognitive task was found, as shown with the assessment of changes in separate tests and using *p*-value synthesis for the overall effect on WM. These findings provide additional evidence for the state-dependency of rTMS effects. The underlying mechanism of this state-dependency might be the homeostatic plasticity that maintains excitation–inhibition balance, which might be necessary for preventing the overload of WM-related neural circuits [42]. The homeostatic inhibition can play an important role both in healthy volunteers as well as in patients with WM impairments, which should be considered by further implementation of combined NIBS protocols into clinical practice. Moreover, the changes in the high-load spatial n-back task after the TMS + WM − were significantly different from rTMS applied during the maintenance period of the Sternberg task. Although exact mechanisms of this state-dependency are poorly understood, the combination of rTMS with cognitive tasks should be carefully investigated and a possible negative influence of concurrent cognitive activity on rTMS effects should be taken into consideration. 

In our study, we have also not found any significant difference between the influence of HF rTMS protocols applied during different periods of the Sternberg task (TMS + WM + (maintenance) and TMS + WM + (rest)), which is consistent with the results of a recently published study investigating the online-effects of HF rTMS [33]. One of the possible explanations for these results might be our lack of knowledge about the temporal resolution of HF rTMS effects. Despite the assumed transience of the single HF rTMS train effects used in most studies, it has not been directly proven yet [43]. Therefore, it is possible that the beginning and duration of the effect of a single train might exceed the stimulation time and match other periods of the investigated cognitive task. Moreover, the duration of HF rTMS trains and ITIs might also influence their effects, considering their relatively uninvestigated role in TMS mechanisms [32]. It should also be noted that the effects of a single train might differ from the effects of several trains leading to the cumulative effects of rTMS during a single session and increasing the complexity of interpreting the online-effects of rTMS. For offline-effects, cumulation has also been proposed due to the long-term effects of rTMS lasting more than the session itself (post-session effects) [43]. However, the impact of post-session effects in the present study is unlikely because of an interval of no less than a week between sessions. Nevertheless, cumulative effects should be taken into consideration during the interpretation of results of rTMS studies with several successive sessions of a protocol, because the measured effects may be caused by a combination of several sessions and not explainable by the latest session only.

Another possible explanation for the non-significant effects of combined protocols can be the high intensity (80%–100% of rMT) of HF rTMS protocols used in our study, which might contribute to the state-dependency of rTMS effects, as proposed by Silvanto et al. (2017) for online-TMS-approaches [44]. According to their model, high-intensity stimulation might lead to inhibition of both task-relevant and irrelevant (noisy) neural activity, diminishing the resulting rTMS effects compared to mid- and low-intensity rTMS [45]. Chung et al. (2018) have also shown that only mid-intensity, but not low- or high-intensity stimulation over the left DLPFC, demonstrated a significant effect on TMS-evoked potentials measured by TMS-EEG [18].

Consistent with our results, significant effects of HF rTMS protocols without a cognitive load on WM performance have been obtained in several studies. Different TMS protocols were used, including HF [46] and LF rTMS [19], as well as cTBS [21,47] and iTBS [20,48]. Improvement of WM accuracy and reaction times after HF rTMS over DLPFC was also confirmed in a meta-analysis; however, the inhomogeneity of the analyzed studies remains an important limitation [25].

Oppositely directed influence of a protocol of HF rTMS without cognitive activity on accuracy in the high-load WM task with spatial stimuli and span length in SSP found in our study is similar to the results of a study performed by Viejo-Sobeira et al. (2017), where a decrease in accuracy in the high-load verbal n-back task and an increase in span length in the digits backward task after both iTBS and cTBS over the left DLPFC, but not sham, were observed [22]. They explain the observed difference after stimulation of the same area both with an inhibitory (cTBS) and an excitatory (iTBS) protocol in terms of the ability of rTMS to induce noisy neural activity, which is non-specific for certain protocols [22]. It is proposed that limited levels of such noise can facilitate ongoing cognitive activity. This effect is cognitive-load-dependent, and, therefore, it can improve the performance of easy tasks (such as the digit span task) but interfere with more complex cognitive activity (e.g., high-load n-back). In contrast to this study, showing significant effects of TBS on verbal WM; in our study, we observed differences only in WM tasks with spatial stimuli. A possible explanation could be the difference in areas of DLPFC used as stimulation targets in both studies because, in the present study, we applied an fMRI-guided approach to determine the stimulated area.

Along with the concept of rTMS effects due to an increase in so-called stochastic noise leading to improvement in tasks with moderate difficulty level, there are also other possible mechanisms of NIBS effects, which could explain the oppositely directed influence of DLPFC stimulation on different WM tasks. According to the net zero-sum concept, a functional gain is always accompanied by a functional loss leading to a resulting zero-sum of NIBS effects [49]. Considering that different WM tasks are testing different WM components, the gain in spatial maintenance measured by the SSP task is accompanied by a loss in manipulation measured by n-back task leading to resulting zero-sum of the effects of DLPFC stimulation. Moreover, in this concept, the interindividual variability of NIBS cognitive effects is explained by individual variability of capacity limits of cognitive functions that might be the reason for the absence of significant stimulation effects calculated for the group. 

In our study, we determined the stimulation area using fMRI with the modified Sternberg task where the storage of verbal stimuli plays a more important role than the manipulation but found the effect of HF rTMS on both manipulation and storage of information. However, we have found an increase only in SSP—a storage-based task—performance, confirming a distinct role of DLPFC in WM components. Effects of rTMS shown in the present study are consistent with the results of a study investigating online-effects of HF rTMS, where stimulation of the DLPFC area, determined by fMRI data of activation during a manipulation task, has led to disruption of manipulation, but fMRI-guided stimulation of superior parietal lobule affected both manipulation and short-term retention [50]. According to these results, the specific role of DLPFC in different components of WM might also be the underlying mechanism of the “trade-off” between manipulation and storage. 

Otherwise, the decrease in performance of the most difficult WM task in our study—high-load n-back, which probably requires more WM resources—could also result not from activation of a storage-related DLPFC area, but from temporary inhibition of an area, providing executive control upon manipulation and update of information in WM. It is consistent with the hypothesis that the role of DLPFC is not the storage of information in WM itself, but manipulation upon it and control of its retention in other brain areas [6]. It should also be noted that the difficulty of this task is higher than in standard n-back tasks used in most cognitive studies due to simultaneous presentation of verbal and spatial stimuli. Moreover, the disruption of DLPFC activity by HF rTMS might impair the processes of divided attention [51]. In the state of temporary inhibition of DLPFC activity, the role of storage-related brain areas, activated during WM tasks, can become more prominent, leading to the improvement in the storage-based SSP task that was observed in the present study.

In this study, cognitive effects were assessed with the tasks that differ from the task used for the determination of stimulation target and combination with rTMS in order to diminish the learning effect during a stimulation session, which could contribute to its cognitive effects. However, the influence of rTMS over the DLPFC area, activated during a verbal storage-based task such as the Sternberg task, on the performance in other WM tasks supports the hypothesis of its crucial role in WM processes, which is not specific for verbal and spatial stimuli, as previously shown in fMRI studies [52]. Consistent with our data are the results of a study of online-effects, where rTMS over the left DLPFC has led to the disruption of spatial stimuli processing in n-back tasks. Moreover, in this study, lateralized activation of DLPFC played an important role only in the case of simultaneous manipulation upon two types of stimuli, similar to the n-back task in our study [53]. However, in order to confirm the proposal of the stimuli-non-specific role of the left DLPFC in WM, the effect of rTMS on a task assessing the storage of verbal information should be determined in further studies.

No serious AEs were reported, confirming the safety of combined rTMS protocols with simultaneous cognitive activity used in our study for cognitive investigations in healthy volunteers, which is important because of limited data regarding the safety of TMS protocols during a brain activity [36]. According to our results, the most frequent AEs were mild and did not lead to cessation of participation in the study, which is consistent with previous studies assessing the safety of TMS [54]. In this study, we have also found that headache is one of the most common AEs occurring within 24 h after a stimulation session (9.09%, or 4 cases of *n* = 44 sessions). Because this AE has an influence on the rTMS tolerability and might lead to the cessation of stimulation, it should be taken into consideration in further studies.

### Limitations

The first limitation of our study is the cross-over design that might have led to the confounding of the protocol effects by the session order effects. However, an important advantage of this design compared to assigning a separate group to each protocol is that the impact of individual variability on the test results is smaller in the crossover design. A small group size (*n* = 11) is also a limitation of the present study. An estimate of the power of the one-sample Wilcoxon test (used for inferring the effects of individual rTMS protocols and post-hoc comparisons between them) for our sample size was obtained using a simulation with 10^5^ tests under a normal shift alternative. For an effect size of 1 (mean divided by the standard deviation), the calculated power was approximately 0.81. In future studies, we plan to use larger samples to validate our findings and investigate more subtle effects and differences between combined and non-combined protocols.

Secondly, we used an individual fMRI-guided approach to determine the stimulation area in this study, which might also have contributed to the inhomogeneity of our results due to different mental strategies that could be used by participants performing the Sternberg task during the scanning (for example, verbal and non-verbal strategies of remembering the letters). Therefore, stimulated areas within the left DLPFC might have different functions depending on participants’ strategies, which can influence the effect of stimulation in individuals measured by cognitive tests. However, an advantage of the fMRI-based approach is the stimulation of functionally relevant brain areas, which might differ across participants, that is not possible by using “anatomical” approaches of target determination, such as structural MRI.

Moreover, in our study, the task combined with stimulation to change DLPFC activity state was different from the tasks used for assessment of rTMS effect, based on the assumption that DLPFC activity plays a control role in WM independent on a task type. However, it might diminish the observed effects of stimulation combined with cognitive activity. Furthermore, the aim of our study was not to determine the exact neuropsychological mechanisms underlying performance of a particular WM task; therefore, a certain level of WM mechanisms generalization and, particularly, control role of the DLPFC in all WM tasks types, might be appropriate. Despite this explicit limitation, the use of different task types allowed us to diminish the learning effect and to avoid tiredness of the participants after a stimulation session, which could have an impact on the post-stimulation test performance. 

It should also be noted that the motor threshold used in our study has limited accuracy as a technique for determination of intensity in cognitive TMS studies because it does not provide exact information about the excitability of non-motor brain areas [55]. However, due to the lack of other reliable methods, it remains the most widely used approach to determine stimulation intensity in cognitive studies.

## 5. Conclusions

According to our results, the combination of stimulation with a cognitive task may diminish its effects in comparison to HF rTMS without a cognitive load. Therefore, a careful investigation of rTMS combined protocols and cognitive tasks is required in further studies before the use of combined techniques as WM enhancement method in both patients and healthy volunteers. Moreover, in the present study, we have found an oppositely directed influence of stimulation over the left DLPFC without a cognitive load on different components of spatial WM. However, for the explanation of mechanisms underlying state-dependency of HF rTMS, our results should be reproduced in studies performed on bigger samples and using other WM tasks for the assessment of the stimulation effects. Another direction for further investigation is the determination of multiple rTMS session effects on WM. Different effects of rTMS on WM performance might be found in such studies compared to the effects of a single session due to possible cumulative effects of rTMS, which might enable their use as effective treatment approaches.

## Figures and Tables

**Figure 1 brainsci-10-00083-f001:**
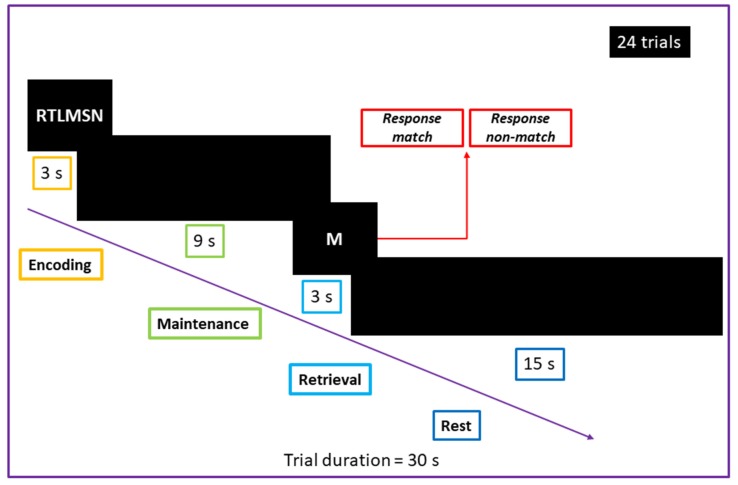
fMRI paradigm with the modified Sternberg task used for individualized determination of stimulation target.

**Figure 2 brainsci-10-00083-f002:**
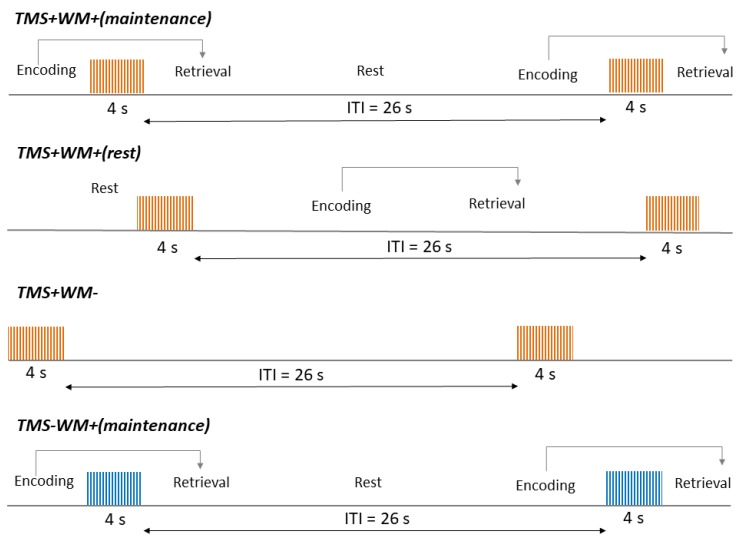
Illustration of high-frequency repetitive transcranial magnetic stimulation (HF rTMS) protocols used in the study. ITI, intertrain interval, TMS + WM + (maintenance), HF rTMS over the left DLPFC during maintenance period of the Sternberg task; TMS + WM + (rest), HF rTMS over the left DLPFC in the rest period between the presentations of the Sternberg task; TMS + WM −, HF rTMS over the left DLPFC without a cognitive load; TMS − WM + (maintenance), HF rTMS over the vertex during maintenance period of the Sternberg task.

**Figure 3 brainsci-10-00083-f003:**
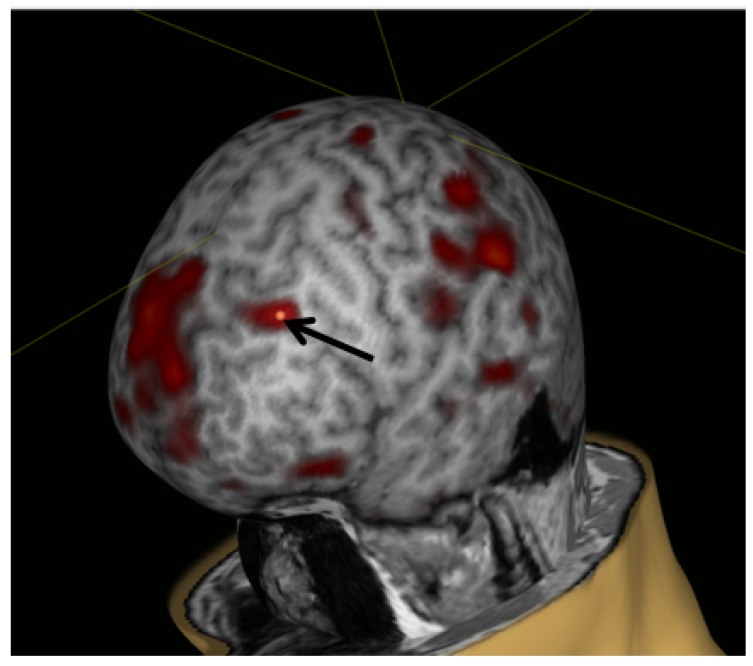
Example of the stimulation target for rTMS over left DLPFC in a healthy volunteer.

**Table 1 brainsci-10-00083-t001:** Cognitive test scores before and after rTMS.

Test		Score	TMS + WM + (Maintenance) ^1^	TMS + WM + (Rest)		TMS + WM −	TMS – WM + (Maintenance)
Name	*n*	Type					
n-back	2	verbal	Before	1.90 (1.50; 2.43)	2.23 (1.22; 2.43)	2.43 (1.53; 2.85)	2.23 (1.07; 2.77)
			After	2.43 (1.90; 2.84)	1.50 (1.04; 2.77)	2.23 (1.50; 2.77)	2.04 (1.07; 2.77)
			Delta	0.02 (0.00; 0.75)	−0.03 (−0.83; 0.34)	−0.08 (−0.90; 0.34)	−0.42 (−0.87; 1.15)
		spatial	Before	1.80 (1.47; 2.43)	1.90 (1.04; 2.85)	2.45 (1.80; 3.19)	1.90 (1.50; 2.85)
			After	2.23 (1.90; 2.77)	2.23 (1.47; 2.85)	2.43 (1.47; 2.77)	2.23 (1.80; 2.77)
			Delta	0.00 (−0.30; 1.72)	0.13 (−0.34; 0.195)	0.00 (−0.97; 0.63)	0.00 (−0.75; 0.97)
	3	verbal	Before	1.37 (0.10; 1.90)	1.37 (0.64; 1.50)	1.22 (0.79; 2.23)	1.04 (0.37; 2.43)
			After	1.00 (0.57; 1.80)	0.64 (0.14; 1.80)	0.84 (0.14; 1.47)	1.47 (0.84; 1.50)
			Delta	0.00 (−0.50; 0.54)	−0.43 (−0.93; 0.43)	−0.54 (−1.19; 0.00)	0.00 (−0.43; 1.40)
		spatial	Before	1.07 (0.50; 1.37)	1.07 (0.57; 1.80)	1.47 (0.64; 1.90)	1.80 (0.79; 1.90)
			After	1.50 (0.64; 2.23)	1.80 (0.64; 2.04)	0.50 (−0.07; 1.47)	1.07 (0.64; 1.80)
			Delta	0.40 (−0.40; 1.40)	0.40 (−0.40; 1.14)	−0.54 (−1.53; 0.03)	−0.14 (−0.77; 0.27)
SSP		Before	8 (7; 9)	8 (7; 8)		8 (6; 8)	8 (7; 8)
		After	8 (6; 9)	8 (5; 8)		8 (8; 8)	8 (8; 9)
		Delta	0 (−1; 1)	0 (−2; 1)		1 (1; 2)	1 (0; 2)

^1^ TMS + WM + (maintenance), HF rTMS over the left DLPFC during maintenance period of the Sternberg task; TMS + WM + (rest), HF rTMS over the left DLPFC in the rest period between the presentations of the Sternberg task; TMS + WM −, HF rTMS over the left DLPFC without a cognitive load; TMS – WM + (maintenance), HF rTMS over the vertex during maintenance period of the Sternberg task.

**Table 2 brainsci-10-00083-t002:** Differences between test scores after HF rTMS protocols (Wilcoxon test) and the overall effect of each protocol (Fisher’s *p*-value synthesis).

Test			TMS + WM + (Maintenance) ^1^	TMS + WM + (Rest)	TMS + WM −	TMS − WM + (Maintenance)
**Name**	***n***	**Type**				
n-back	2	verbal	*p* = 0.093	*p* = 0.575	*p* = 0.646	*p* = 0.790
		spatial	*p* = 0.241	*p* = 0.415	*p* = 0.475	*p* = 0.674
	3	verbal	*p* = 0.953	*p* = 0.350	*p* = 0.086	*p* = 0.445
		spatial	*p* = 0.139	*p* = 0.386	*p* = 0.045	*p* = 0.374
SSP			*p* = 0.726	*p* = 0.799	*p* = 0.040	*p* = 0.185
Overall effect ^2^			*p* = 0.27	*p* = 0.70	*p* = 0.03	*p* = 0.61

^1^ TMS + WM + (maintenance), HF rTMS over the left DLPFC during maintenance period of the Sternberg task; TMS + WM + (rest), HF rTMS over the left DLPFC in the rest period between the presentations of the Sternberg task; TMS + WM −, HF rTMS over the left DLPFC without a cognitive load; TMS – WM + (maintenance), HF rTMS over the vertex during maintenance period of the Sternberg task. ^2^ Overall effect—significance of the effect of each protocol on any of the WM tests according to the results of Fisher’s *p*-value synthesis.

**Table 3 brainsci-10-00083-t003:** Post-hoc analysis of differences in high-load n-back task with spatial stimuli (Wilcoxon test with Bonferroni correction for multiple comparisons).

Comparison	TMS + WM + (m) vs. TMS + WM + (r) ^1^	TMS + WM + (m) vs. TMS + WM −	TMS + WM + (m) vs. TMS − WM + (m)	TMS + WM + (r) vs. TMS + WM −	TMS + WM + (r) vs. TMS − WM + (m)	TMS + WM − vs. TMS − WM + (m)
*p*	1.000	0.048	0.66	0.372	1.000	1.000

^1^ TMS + WM + (m), HF rTMS over the left DLPFC during maintenance period of the Sternberg task; TMS + WM + (r), HF rTMS over the left DLPFC in the rest period between the presentations of the Sternberg task; TMS + WM −, HF rTMS over the left DLPFC without a cognitive load; TMS – WM + (m), HF rTMS over the vertex during maintenance period of the Sternberg task.

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
