# Peer review of "Combining HF rTMS over the Left DLPFC with Concurrent Cognitive Activity for the Offline Modulation of Working Memory in Healthy Volunteers: A Proof-of-Concept Study"

_brainsci, 2020, doi:10.3390/brainsci10020083_

Round 1

Reviewer 1 Report

The proposed manuscript investigates the question of whether the effectiveness of rTMS protocols as cognitive enhancement technique can be increased by combining the stimulation with concurrent cognitive activity.

A thorough investigation of this question with the appropriate methodological approach could be of interest to the field, also in order to further understand with which mechanisms TMS actually changes brain processes. Unfortunately, the presented study does not provide convincing evidence to rule out the effectiveness of the combined rTMS method in their case. This is in my opinion due to the small sample size N = 11) as well as due to the choice of the Sternberg task as cognitive task to activate the to-be-stimulated brain region in the dlPFC – which has divergent demands on the WM system from the n-back and spatial span task.

I detail my concerns in the following:

The test for differences in the respective WM tasks pre-vs. post-stimulation might be appropriate, but the sample size in the chosen design is so small that checking the assumptions of the test is difficult or impossible. Given the many comparisons calculated, the two “significant” effects (which also go in opposite directions) seem likely to be false positives. Why was the sample size chosen to be 11 (12) subjects? In an extension to this: the presented design has no more than 0.78 power to detect effect sizes as large as 0.8. It seems like the authors would want to detect an effect if it were indeed that large. Is their evidence to assume, that the effect of non-combined vs. combined protocol and also the effect of target TMS vs. vertex TMS is that large?

In order to make the claim, that there were no differences between the pre- and post-stimulation task performances I further advise to use Bayesian statistics, to calculate evidence for the null – this way the author could also quantify that their data simply does not contain enough evidence for either a difference nor the null. The results are presented in a very non-accessible way in Table 1. On top of just showing a lot of numbers, the format is shifted in some cases. I recommend to present the data in plots and to highlight the significant comparisons. I am concerned, that the choice of WM task for dlPFC-activity localization actually prevented the authors to find stimulation effects in the combined protocol. The Sternberg task, especially when the stimuli are presented simultaneously and only item, but not binding memory is required at test, don’t put the same demands on the WM system as the n-back as well as the spatial span task. In the latter, the system is required to bind up to 6 memoranda to their serial (verbal n-back) and their serial as well as spatial location (spatial n-back). The latter is the case also for the SSP. Thereby, the region in the dlPFC localized in the Sternberg-task, might not be the most activated part of the brain in tasks actually challenging the working memory system (see Oberauer, K. (2019). Working Memory Capacity Limits Memory for Bindings. Journal of Cognition, 2(1).)

Overall the terms used to describe the Sternberg-Task are rather unusual. In Figure 1 e.g. N=24 actually refers to the (actually also way to small) number of trials, although N normally refers to sample size. Further, the authors state “The task consisted of four blocks: encoding, maintenance, retrieval, and rest”. These are phases within the task not blocks, which more commonly refer to a number of trials. Rather a trial consists of four phases.

Reviewer 2 Report

Here are my comments.

Reviewer 3 Report

This is an interesting and novel study that explores the effects of combination of DLPFC rTMS and cognitive activation on working memory. Interestingly, authors found a negative effect of combination with Greater function enhancing by rTMS alone.

This effect is certain due to state dependent mechanisms among which homeastic inhibition due to overloading activaction, principally described for motor task and in diseases with increased cortical excitability, could perhaps  play a relevant role.

This point I think should more stressed in the discussion.

Generalli the study is really fine: methods are sound, results well presented and discussed.

Round 2

Reviewer 1 Report

The authors added the statement "it might be enough to find reasonably large differences in effects of combined and non-combined protocols.” in their manuscript based on my comment on the small sample size. I am still unconvinced by this argument. What do the authors base their claim of "might be enough" on? Are there previous studies with combined protocols finding large effects with this sample size? Where there power calculations?
